# A Review of the Epidemiology of Lassa Fever in Nigeria

**DOI:** 10.3390/microorganisms13061419

**Published:** 2025-06-18

**Authors:** Danny Asogun, Bosede Arogundade, Faith Unuabonah, Olorunkemi Olugbenro, Joyce Asogun, Fatelyn Aluede, Deborah Ehichioya

**Affiliations:** 1Institute of Viral and Emergent Pathogens, Irrua Specialist Teaching Hospital, Ekpoma 310115, Edo State, Nigeria; joysasogun@gmail.com (J.A.); drfatelyn2017@gmail.com (F.A.); ehichioya@bnitm.de (D.E.); 2Department of Community Medicine, Ambrose Alli University, Ekpoma 310104, Edo State, Nigeria; 3Department of Planning, Research & Statistics, Federal Ministry of Health, Abuja 900211, FCT, Nigeria; docboliza@gmail.com; 4Department of Haematology and Transfusion Science, Ambrose Alli University, Ekpoma 310104, Edo State, Nigeria; faithunuas@gmail.com; 5Department of Microbiology, Faculty of Life Sciences, Ambrose Alli University, Ekpoma 310104, Edo State, Nigeria; oolorunkemi@gmail.com

**Keywords:** Lassa fever virus, epidemiology, outbreaks, surveillance, response

## Abstract

Lassa fever, a viral hemorrhagic illness that first came into the limelight as a clinical entity in 1969 when it was discovered in Northern Nigeria, is now found in other West African countries such as Sierra Leone, Liberia, Guinea, Togo, and the Benin Republic. Over the years, the disease, which is primarily transmitted from contact with infected mastomys rodents to humans, has the capability of secondary human-to-human transmission with significant morbidity and mortality, especially in healthcare settings. The disease is typically characterized by seasonal outbreaks, which peak during the dry season months of December to March. Lassa fever significantly impacts public health and the socioeconomic life of people in affected communities. In Nigeria, the Integrated Disease Surveillance and Response Strategy (IDSR), along with other medical countermeasures, have been employed to curtail the impact of the disease in endemic regions of Nigeria and other West Africa countries. The one-health approach to combat the disease is a promising strategy. This, along with the hope of a safe and effective vaccine, is a ray of hope on the horizon for public health authorities in Nigeria and other West African countries that the battle against Lassa fever might indeed end sooner than later.

## 1. Background

Lassa fever is a significant acute viral hemorrhagic fever caused by a spiked enveloped virus called Lassa virus. It is a member of the arenavirus family of viruses. The virions can be pleomorphic, with diameters ranging from 50 to 300 nm; however, they are typically spherical and average 120 nm in diameter. The genome consists of two single-stranded ambisense RNA segments: large (*L*) and small (*S*) segments. The 3.5 kb *S* segment encodes the glycoprotein precursor (GPC) in the positive sense and nucleoprotein (NP) in the antisense, while 7.2 kb *L* segment encodes the Z protein (RING-finger matrix protein) in the positive sense and an RNA-dependent RNA polymerase (RdRP) called L protein in the antisense [1,2].

Lassa virus was first discovered in Lassa, a town in Nigeria in 1969 [3]. Lassa fever subsequently spread to many parts of the country, with outbreaks reported in most parts over the last 50 decades [4,5]. Human beings can be infected with Lassa fever virus when the rodent vector (*Mastomys natalensis*) contaminates food items and things in the home with its urine or feces [6,7]. High infection rates among rodents contribute significantly to human infections, primarily through exposure to contaminated food, water, or household environments [8,9].

Lassa fever has been of major concern to the public health of most West African countries, particularly in regions with tropical conditions that support rodent vector survival [10,11,12]. The infection can lead to serious complications like widespread intravascular blood clotting, extensive bleeding in mucosal surfaces, multiple organ failure, and hypotension, causing shock with the need for advanced life-saving procedures [13].

Although significant advancements have been made in understanding Lassa fever’s replication patterns, pathogenesis, and molecular epidemiology, and despite the development of cutting-edge diagnostic technologies, efforts to control Lassa fever outbreaks in West Africa remain challenging and ineffective; hence, this is one of the aims of this review.

The annual infection rate of Lassa virus across West Africa is from 100,000 to 300,000 cases, out of which approximately 5000 deaths result [14]. It is estimated that out of every 100 persons who get Lassa fever, 1 person may die, but for those who are hospitalized, the risk of death can be as high as 15 in every 100 persons [15]. Unfortunately, such estimates are crude, because surveillance for cases of the disease is not uniformly performed [15]. Lassa fever carries with it a considerable socioeconomic burden on health resources and economic stability throughout Sub-Saharan Africa. This impact raises alarming concerns among health officials tasked with managing outbreaks and patient care within resource-constrained settings [10,12].

The epidemiology of Lassa fever integrates factors such as geographic distribution, host–pathogen dynamics, and socio-political contexts [16,17]. In Nigeria, phylogenomic studies have demonstrated the persistence of multiple viral lineages, highlighting the complexity of disease transmission patterns [18]. Despite the significant public health impact of the disease, Lassa fever remains underreported in many endemic regions due to limited surveillance infrastructure, insufficient diagnostic capabilities, and healthcare workforce shortages [17,19].

Since Lassa virus’ first discovery [20], substantial outbreaks that transpired between 1969 and 1978 in Nigeria, Sierra Leone, and Liberia played a pivotal role in improving our understanding of the disease [10]. These outbreaks led to the development and enhancement of both surveillance activities and response strategies within affected communities. For instance, the alarming fatality rates observed during the 1969 outbreak, which even included fatalities among healthcare workers, brought about the implementation of stringent infection control measures within various medical facilities to contain the spread [3,11].

This study provides an overview of the trends in reference to the geo-spatial spread and increasing number of confirmed Lassa fever cases and highlights effort by the national public health institute, Nigeria Centre for Disease Control (NCDC), to respond to the seasonal outbreak of Lassa fever in Nigeria. Specifically, this paper attempts to draw the attention of public health experts, epidemiologists, researchers, and other stakeholders engaged in the control of Lassa fever to the need to focus on the reasons behind the expansion of the epidemic despite efforts at control. Against this backdrop, surveillance approaches and control measures of Lassa fever are highlighted as carried out within the framework of the Integrated Disease Surveillance and Response (IDSR) framework. The establishment of the Nigeria Centre for Disease Control (NCDC) and the adoption of the IDSR framework have significantly enhanced the country’s capacity to detect outbreaks, monitor trends, and facilitate rapid responses in recent years. 

While earlier research addresses similar epidemiology themes, our study introduces new elements we believe contribute to advancing the discourse in this area. Earlier authors with similar goals such as Okwor et al. (2018) [21] describe the epidemiological trends of Lassa fever in Nigeria from 1969 to 2017, while Dalhat et al. (2022) [22] extended the analysis to the period from 2018 to 2022. Our study expands the timeline to include data up to 2024 and emphasizes the roles and approaches of the Integrated Disease Surveillance and Response (IDSR) team in early detection, reporting, analysis, and control across different levels of Nigeria’s healthcare system. This provides valuable insight into how the Nigeria Centre for Disease Control (NCDC) employs the IDSR framework to coordinate responses to Lassa fever outbreaks. Additionally, our review highlights the critical importance of interdisciplinary research as guided by the one-health concept.

## 2. Geographical Distribution

Seasonal fluctuations exert a significant influence over the incidence of Lassa fever, with many outbreaks observed to occur simultaneously with climate shifts, particularly during the rainy season when environmental factors are conducive to increased rodent populations and subsequent human exposure to the infecting virus [23]. The disease is enzootic among some rodent and non-rodent populations in certain regions of West Africa [24,25]. Benin, Ghana, Guinea, Liberia, Mali, Sierra Leone, and Nigeria are known to be endemic for Lassa fever. Other West African nations might also have the disease, but this is yet to be confirmed due to limited surveillance data [11,25,26]. The virus resides mainly within the natural habitat of *Mastomys natalensis*, its primary reservoir [16,19].

Within Sierra Leone, Lassa fever has become widespread, while in Nigeria, specific urban areas within states like Edo and Ebonyi are identified as high-risk zones that require increased disease surveillance [10,27]. The geographical distribution of Lassa fever is influenced by a variety of factors, such as local climate conditions, seasonal changes, and the prevalence of the multimammate rat, which plays a crucial role in the virus’s transmission [11,20].

Understanding the correlation between the dynamics of rodent populations and various environmental factors is key for developing targeted epidemiological studies. These studies possess the potential to inform decisive public health actions against the disease [10,26].

The virus has undergone substantial genetic divergence over time, resulting in the presence of seven phylogeographic lineages (I–VII) in West Africa (Figure 1). Lineages I, II, and III are found in Nigeria; the first lineage in the north-eastern part of the country with the remaining two commonly found in the southern and north-central regions, respectively [28,29].

## 3. Lassa Fever Incidence, Cases, and Trends in Nigeria

From 1969 to date, there has been an increase in Lassa fever outbreaks characterized by an increase in its incidence and geographical spread. Initially, outbreaks mainly occurred in northern Nigeria, with a few cases in the eastern states, but 1989 heralded outbreaks in the south–south zone, with highest number of outbreaks currently recorded in Edo state. Between 1969 and 1989, only 1–4 states reported outbreaks; from 1996 onwards, the number of states increased to nearly one-third of the country until 2009, after which there has been notable and steady increase in the number of affected states, with outbreaks reported in all of the six geopolitical zones. By 2020, only FCT and Bayelsa have not reported any suspected Lassa fever case [5,21]. Figure 2 reveals a steady rise in suspected Lassa fever cases from 2018 to 2023, except the year 2021, where there seems to be a decline.

A recent review on Lassa fever cases from 2020 to 2023 stated that the overall suspected cases reported from the 36 states in the country and FCT was 28,780. Out of these, 4036 cases were confirmed positive for Lassa fever in the laboratory, with 762 deaths recorded [31]. This gives an overall case positivity rate of 14%. The year 2020 had the highest case positivity rate (17.5%) compared to 2021, 2022, and 2023, with 11.0%, 13.0%, and 13.9%, respectively [Table 1]. In the year 2024, although the number of states reporting at least one confirmed case of Lassa fever remained 28 states, as it was in the year 2023, few states that initially reported in the preceding year did not and the case positivity rate reduced to 13.0% [Figure 3 and Figure 4].

**Table 1 microorganisms-13-01419-t001:** Suspected and confirmed cases of Lassa fever and their reporting states in Nigeria between 2018 and 2024 [32,33,34,35,36,37,38].

Year	Suspected Cases	Confirmed Cases	Calculated Case Positivity (%)	No. of Death	Calculated CFR (%)	States with at Least 1 Confirmed Case
2018	3498	633	18.1	171	27.0	23 states, 93 LGA
2019	5057	833	16.5	174	20.9	23 states, 86 LGA
2020	6791	1189	17.5	244	20.5	27 states, 131 LGA
2021	4654	511	11.0	102	20.0	17 states, 68 LGA
2022	8202	1067	13.0	189	17.7	27 states, 112 LGA
2023	9155	1270	13.9	227	17.9	28 states, 124 LGA
2024	10,098	1309	13.0	214	16.3	28 states, 139 LGA

**Figure 2 microorganisms-13-01419-f002:**
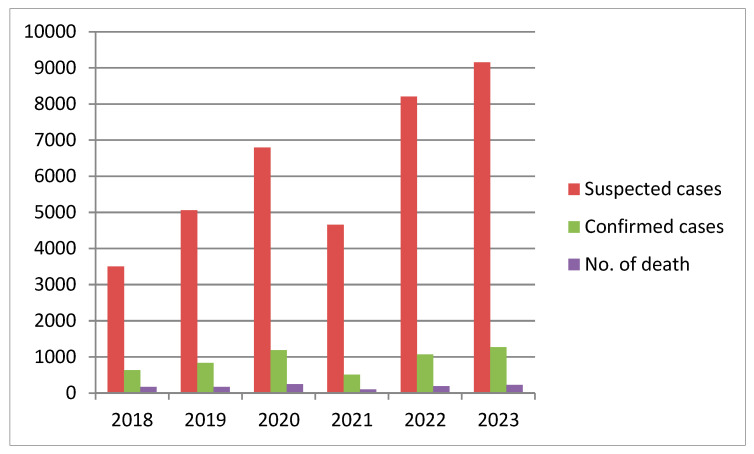
Trends of suspected cases, confirmed cases, and death reported between 2018 and 2023.

**Figure 3 microorganisms-13-01419-f003:**
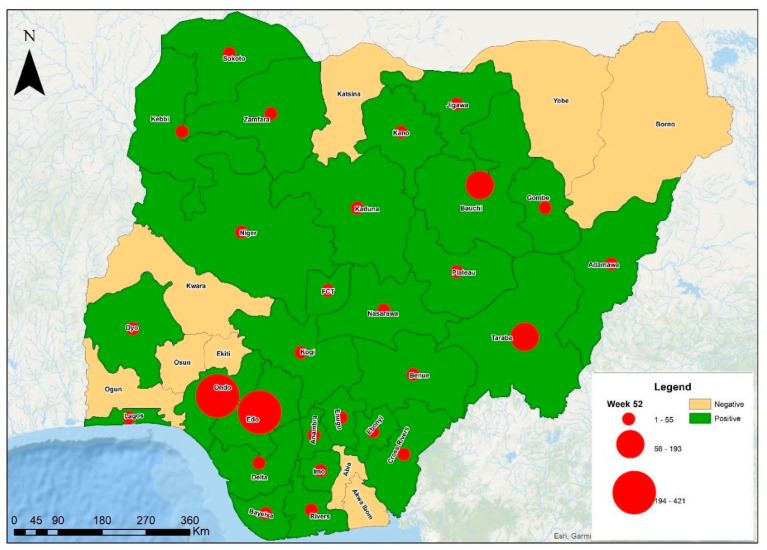
The prevalence of confirmed Lassa fever cases by state in Nigeria, year 2023 [38], displaying the cumulative number of cases in each state from Week 1 to Week 52 (annual).

**Figure 4 microorganisms-13-01419-f004:**
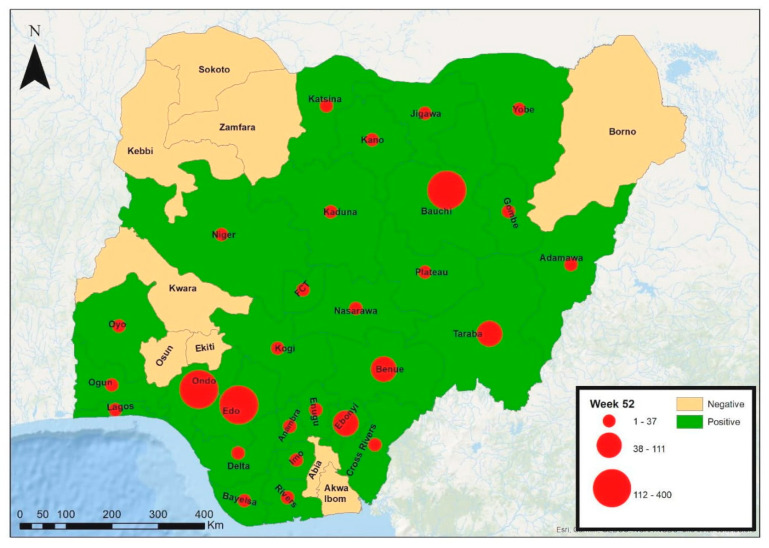
The prevalence of confirmed Lassa fever cases across states in Nigeria, year 2024 [32], displaying the cumulative number of cases in each state from Week 1 to Week 52 (annual).

The male to female ratio from 2018 to date has been predominantly 1:0.9 (2020, 2021, and 2023), falling to 1:0.8 in 2022 and rising to 1:1.1 and 1:1 in 2019 and in 2024, respectively. Within these reviewed years, the predominant age group is 21–30 years and the median age ranges from 29 to 34. Ondo, Edo, Ebonyi, and Bauchi are major states in reporting confirmed cases in these years. In 2024, 72% of confirmed cases were reported from Ondo, Edo, and Bauchi (31%, 22%, and 19%, respectively); in 2023, 76% were reported from Ondo, Edo, and Bauchi (34%, 27%, and 15%); in 2022, 72% were reported from Ondo, Edo, and Bauchi (33%, 25%, and 14%); in 2021, 84% were reported from Edo, Ondo, and Bauchi (42%, 34%, and 8%); in 2020, 75% were reported from Ondo, Edo, and Ebonyi (36%, 32%, and 7%); in 2019, 71% were reported from Edo, Ondo, and Ebonyi, (37%, 34%); and in 2018, 80% were reported from Edo, Ondo, and Ebonyi (44%, 25%, and 11%) [32,33,34,35,36,37,38].

Nigeria, with a current population of about 234,852,000 [39], has year-round reporting of Lassa fever cases and the incidence of fever peaks between the last quarter of one year to the first quarter of the next year, mostly November to March (dry season), with low incidence in the rainy season [22,40].

## 4. Seasonal Variation

Currently, Lassa fever is endemic in 16 countries, with seasonal outbreaks correlating with increased rodent–human contact during the dry season [41]. The prevalence of Lassa fever is intricately tied to environmental fluctuations impacting West Africa, accentuating the ecological dimensions linked to this zoonotic disease [10,42]. Among the influential factors are rainfall levels, humidity, temperature variations, and changes in day length, all of which profoundly affect transmission dynamics [11,25]. Outbreaks of Lassa fever typically align closely with the onset of the rainy season, which creates favorable conditions for increased rodent populations, thereby raising the chances of human exposure to the virus [29]. Local observations and studies conducted in Guinea suggest that the timing of Lassa fever outbreaks coincides with the germination of primary food sources for rodents, leading to population surges in their numbers within farming regions [26]. This cyclical relationship between environmental conditions and the patterns of disease transmission accentuates the critical necessity for ongoing research and meticulous surveillance aimed at deepening our understanding of the dynamic trends associated with Lassa fever [10,20].

## 5. Risk Factors

Lassa virus infection and transmission are influenced by various risk factors, which contribute to outbreaks and their endemic nature in certain regions. Primary infection with LASV occurs when individuals come into contact with droppings or urine of the reservoir host, *Mastomys natalensis*, or surfaces contaminated with them [6,22,43]. Individuals at high risk include those residing, working, or spending extended periods in areas where reservoir rodents are present and scavenge on human food remains or improperly stored food [20,44].

Rodent infestation has been linked to poor environmental sanitation, such as inappropriate disposal of household waste, overgrown bushes surrounding residential buildings, and sun-drying food items in the open, accessible to rodents [45,46,47]. Studies have highlighted that households with a high occurrence of Mastomys trapped had the poorest housing conditions and hygiene practices [48,49,50] and at times corresponding Lassa seroprevalence in humans [8]. In a Sierra Leonean refugee camp, Bonner et al. assessed how the quality and hygiene conditions of households can be possible risk factors for rodent infestation and infection with Lassa fever. They reported that the houses of those infected had the poorest housing conditions and external hygiene, and the odds of having a rat burrow were higher in the houses of those infected than in control houses [51].

Human behavior and practices that enhance human interactions with zoonotic hosts may expose them to a level of risk of viral infection [52]. In certain communities, hunting and trapping animals contribute to an increased risk of Lassa fever transmission. Mastomys rodents, which are sometimes consumed as food, present a direct transmission risk during handling and preparation due to the high viral load in their blood. Activities such as farming, bush burning, and deforestation significantly increase the likelihood of contact between humans and rodents [23,53,54]. Farmers often encounter rodents on their farms during intensive agricultural activities and while bringing harvested produce into their homes, creating sustained contact with the reservoirs. Within households, consuming food contaminated by rodent feces is considered a potential source of infection [44,55,56]. Future changes, including population growth, urbanization, land use patterns, and climate change, are expected to exacerbate the frequency and seasonality of zoonotic disease outbreaks, including Lassa fever [57,58,59].

Transmission from one individual to another can result from direct contact with infected blood, urine, or pharyngeal secretions. As early symptoms of LF are frequently mild and often mimic other febrile illnesses, Lassa fever diagnosis may be challenging and hence it may remain undetected in households and within the community [60,61]. In healthcare settings, nosocomial transmission has highlighted critical gaps in infection prevention and control measures. Healthcare workers, when ill-equipped with personal protective equipment (PPE), are especially vulnerable and face substantial risks. Lassa fever has been transmitted from patients to hospital staff, leading to high mortality [62,63,64,65]. These gaps have now been mitigated by robust training and resource allocation to control protocols, thereby significantly reducing the spread of infection to healthcare providers.

Insufficient education and awareness about Lassa fever have increased individual risk and delayed responses, resulting in adverse health outcomes. Awareness campaigns and education are crucial for effective prevention of LF in rural and less educated populations [66,67]. A study compared the knowledge of Lassa fever between a community in Ibadan recently affected by the disease and a nearby university community with no history of outbreak. The study revealed that despite the presence of similar healthcare access in the two communities, the community with the recent outbreak had significantly less knowledge of Lassa fever reservoirs, its other means of transmission, and necessary preventive measures [68].

Rodent control remains key to LF prevention, with housing improvements and sanitation as vital strategies to reduce human–rodent contact. Addressing these ecological, socioeconomic, and public health challenges requires an integrated, interdisciplinary, and concerted approach that must be sustained to reduce disease burden and improve outcomes in endemic regions.

## 6. Surveillance of Lassa Fever in Nigeria

Surveillance is an essential disease control program that can be achieved passively (when health facilities or the community submit routine reports) or actively (when cases are searched out by regularly contacting health workers on the phone or visiting the community) or via Integrated Disease Surveillance (IDSR). The WHO AFRO member states endorsed the IDSR in 1998, which was subsequently adopted in Nigeria in the year 2001 in order to improve disease detection and obtain prompt responses to priority diseases. Epidemic-prone diseases like Lassa fever, yellow fever, measles, cerebrospinalmeningitis, and other diseases of public health importance have been listed among the priority diseases under IDSR’s umbrella. The IDSR uses a multi-pronged approach combining structured data from indicator-based surveillance and information captured as alerts in event-based surveillance methods in the active case search of Lassa fever during outbreaks [69].

Although conceived in 2007, the first official step into the establishment of the Nigeria Centre of Disease Control (NCDC) was in 2011, when the Nigeria Field Epidemiology and Laboratory Training Programme (NFELTP) and some arms in the Federal Ministry of Health formed the foundation of the agency in response to some disease outbreaks that occurred in the previous years [70]. In 2018, the Bill for an Act for its establishment was eventually signed into law by the then president [71], and the agency has been a major player in strengthening the surveillance of Lassa fever and other infectious diseases. It monitors the trend of diseases nationwide, coordinates the country’s preparedness to respond promptly during outbreaks, and supports affected states with the resources and expertise needed [71,72]. The IDSR framework and the establishment of the NCDC have enhanced the country’s ability to monitor trends and detect outbreaks by systematically collecting, analyzing, and disseminating data at local, state, and national levels.

One of the departments the agency employs in accomplishing its task is the Surveillance and Epidemiology arm, which collects data on Lassa fever and other priority diseases from all the states in the country using the IDSR strategy. This data, when collated and analyzed, helps in detecting outbreaks and guides decisions on required policies. The department also releases the national *Weekly Epidemiological Report* every Tuesday for each week of the year. The department is also always on the lookout to detect and verify rumors relating to diseases and outbreaks using an event-based surveillance system. In addition to this, the Health Emergency Preparedness and Response department mitigates emergencies and supports responses to outbreaks. For example, this department activated the Lassa Fever Emergency Operations Center (LF-EOC) on 28 January 2023 in response to an unexplainable increase in the number of confirmed cases over previous years and increased records of infection and death of health workers. The Emergency Preparedness and Response Directorate also deployed Rapid Response Teams (RRTs) between January and March 2025 to states with the highest outbreaks: Ondo, Edo, Ebonyi, Bauchi, Benue, and Taraba. Also worthy of note is the Public Health Laboratory department, which provides diagnostic services for Lassa fever and other diseases of public health importance by coordinating national and regional reference laboratories and managing the National Public Health Laboratory Information Management System for the country. This department also deploys mobile laboratories when responding to outbreaks [71,73,74].

Laboratory-based surveillance consists of the collection, analysis, and interpretation of laboratory data to detect and monitor disease spread and emerging pathogens and to ascertain the effectiveness of control measures [75]. The diagnostic capacity for Lassa fever surveillance has improved through the establishment of specialized laboratories, including the National Reference Laboratory and other accredited laboratories that conduct RT-PCR testing for Lassa viruses. These laboratories play a vital role in confirming cases and guiding treatment. Diagnostic laboratories for Lassa fever are in the following centers in the country: NCDC National Reference Laboratory (NRL) in Abuja, Irrua Specialist Teaching Hospital (ISTH) in Edo State, Lagos University Teaching Hospital (LUTH), Alex Ekwueme Federal Teaching Hospital in Ebonyi State, Federal Medical Center Owo in Ondo State, Ondo State Public Health Laboratory in Akure, Chukwuemeka Odimegwu Ojukwu Teaching Hospital (COOUTH) in Akwa, and the first NCDC Zonal Reference Laboratory in Ado-Ekiti [76].

Irrua Specialist Teaching Hospital (ISTH) serves as a primary referral center for Lassa fever and was designated a Centre of Excellence for its management in 2001. Since then, it has been instrumental not only in treating Lassa fever but also in raising awareness and managing other viral hemorrhagic fevers, including COVID-19 and Mpox. The diagnostic laboratory at ISTH was established in 2008. It is domiciled at the Institute of Viral and Emergent Pathogens Control and Research (IVEPCR), formerly known as the Institute of Lassa Fever Research and Control (ILFRC). The institute has played a pivotal role in the clinical management, surveillance, and public health response to viral hemorrhagic fevers. Its contributions have been instrumental in strengthening Nigeria’s preparedness and response to infectious disease outbreaks, thereby enhancing national and regional health security [40].

## 7. Current Control Efforts of Lassa Fever

Since Lassa fever was discovered in Nigeria over 50 years ago, a lot of progress has been made in controlling the disease (Figure 5).

Some of the control efforts are as follows:i.Case Management:

Case management efforts have led to the establishment of specialized Lassa fever treatment centers equipped with isolation wards. These centers are essential for the proper care and containment of the disease. Additionally, the provision of Ribavirin, the antiviral drug used for treatment, is a vital part of the response, with protocols in place for its early administration to enhance its effectiveness.

ii.Infection Prevention and Control (IPC):

Infection prevention and control measures have been implemented, including the use of standard precautions in healthcare facilities and personal protective equipment (PPE). Guidelines for infection prevention and control have been developed, accompanied by training programs for frontline healthcare workers.

iii.Public Health Education and Risk Communication:

Public awareness campaigns emphasize rodent control, food storage practices, and personal hygiene. Community engagement is carried out through radio programs, town hall meetings, and the distribution of educational materials in local languages.

iv.Environmental and Rodent Control:

Efforts to promote environmental and rodent control include advocating for safe food storage practices to reduce rodent access and initiating community-led initiatives for rodent control and improved sanitation.

v.Research and Development:

Research and development efforts are ongoing, with clinical trials and studies focused on Lassa fever vaccines and partnerships with international research institutions aiming to enhance our understanding of the disease and develop innovative control strategies.

## 8. Response Strategies

I.Policy and Governance:

The governance and leadership structure for Lassa fever control consists of a specialized agency of government (NCDC) for disease control, a one-health approach, treatment and isolation centers across selected tertiary facilities in the country, and designated molecular laboratories for diagnostics. There are national guidelines and protocols for Lassa fever management and the integration of Lassa fever control efforts into broader national health security frameworks.

II.Outbreak Preparedness and Emergency Response

Outbreak preparedness and emergency response have been bolstered with the establishment of the National Lassa Fever Emergency Operations Centre (EOC) to ensure a coordinated outbreak response. Rapid response teams are deployed to affected areas, and contingency plans alongside stockpiles of essential medical supplies have been developed.

III.Surveillance and Contact Tracing:

Surveillance and contact tracing are conducted through active case finding and monitoring contacts for 21 days to detect secondary cases. Digital tools (SORMAS) are utilized for real-time data collection and analysis, enhancing the effectiveness of these efforts. Cross-border collaboration has been emphasized, encouraging regional cooperation with neighboring countries for information sharing and coordinated responses.

IV.Capacity Building:

Capacity building is another vital aspect, with regular training and simulation exercises provided for healthcare workers and emergency responders. The laboratory networks have been strengthened to ensure rapid diagnosis and confirmation of cases.

A close link has been observed between the health of humans, animals, and the ecosystem in general, and alterations in this linkage can either increase the risk of humans and animals developing/spreading diseases or be positively utilized in combating emerging infectious diseases. One health is, therefore, a unified approach to optimize these three systems in controlling diseases. From assessment, this weapon has been found to be a potent one in curtailing zoonotic diseases like Ebola, rabies, avian influenza, etc.; vector-borne diseases like malaria, dengue fever, West Nile, etc.; food-associated diseases like salmonella, norovirus, listeria, etc.; antimicrobial resistances; and even environmental health issues like pollution and climate change [78].

Lassa virus, being a zoonotic infection, requires a one-health approach for good control. In Nigeria, use of the one-health approach in tackling Lassa fever has started, although the human health component of the one-health approach has advanced more than environmental and animal health.

## 9. Conclusions

Lassa fever has been identified as a growing health problem in West Africa and it is imperative that the subregion steps up to its responsibilities in combating the threat of Lassa fever that challenges its public health ecosystem. The journey so far for Lassa fever surveillance in Nigeria and other West Africa countries has not been without many hurdles and challenges. Despite the efforts to control the disease, Lassa fever remains underreported due to limited surveillance and infrastructure, insufficient diagnostic capabilities, and healthcare workforce shortage. To effectively combat the disease, a multifaceted approach must be used, one that incorporates a robust contact tracking system and strengthened healthcare systems while addressing ecological, socioeconomic, and cultural factors.

The one-health approach has been advocated as a strategy that must be employed in the control of Lassa fever and other diseases of zoonotic origin with epidemic and pandemic potential [79]. This approach to disease control is a multisectoral strategy that brings to the table the understanding of how humans, animals, and the environment influence the emergence of diseases.

Community education and awareness campaigns to promote behavioral change, research and development of effective vaccines and therapeutic interventions, and collaboration and coordination among stakeholders must all be carried out to ensure a unified response to Lassa fever outbreaks. It is only with sincere and concerted efforts and with team spirit that this goal can be achieved.

Finally, those at the helm of affairs of government, as well as policymakers and politicians, must create an enabling environment for translational research and demonstrate a willingness to fully implement recommendations. Policymakers should support the prioritization and translation of research findings into the development of a national agenda for the control of Lassa fever. Such priorities should include allocating sufficient resources, influencing the enactment of public health laws towards preventive health practices, and putting in place a governance structure for the sustainability of Lassa fever control programs.

## Figures and Tables

**Figure 1 microorganisms-13-01419-f001:**
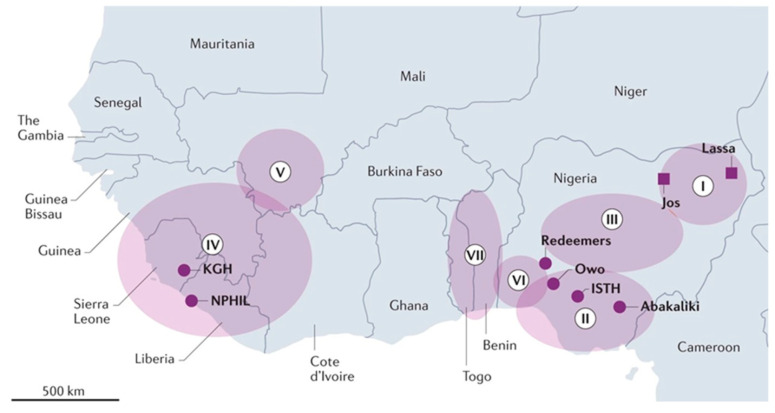
The different Lassa virus lineages (I–VII) in West African countries [30].

**Figure 5 microorganisms-13-01419-f005:**
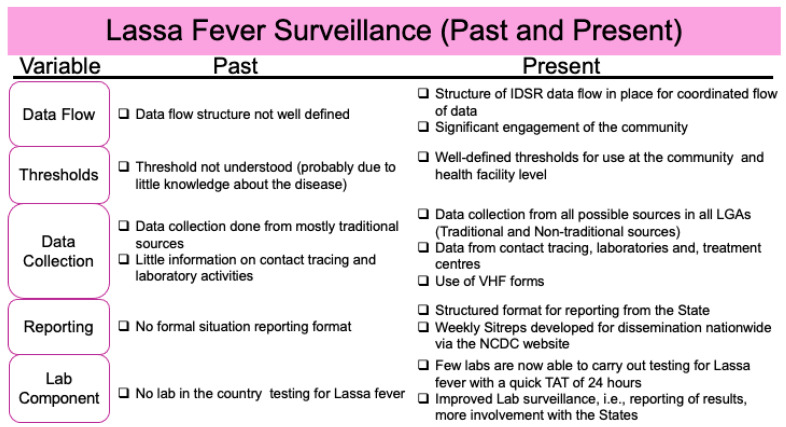
Improvements in Lassa fever surveillance: adapted from [77].

## Data Availability

No new data were created or analyzed in this study.

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
