# Peer review of "A Review of the Epidemiology of Lassa Fever in Nigeria"

_microorganisms, 2025, doi:10.3390/microorganisms13061419_

Round 1

Reviewer 1 Report

Comments and Suggestions for Authors

The manuscript "A review of the epidemiology of Lassa fever in Nigeria" shows an updated review of this infectious disease in Nigeria. The manuscript is overall well-written and organized.
I have some comments in order to improve it.
1.I suggest authors to include a section describing the viral biology of the virus (structure, genome organization, genetic variability, etc).
2.I also recommend authors to add a figure showing the average prevalence of Lassa fever confirmed cases across the different states within Nigeria.
3.The abbreviation CFR should be defined in Table 1. 

Author Response

  1. COMMENT: I suggest authors to include a section describing the viral biology of the virus (structure, genome organization, genetic variability, etc).

RESPONSE: This has been done.

  1. COMMENT: I also recommend authors to add a figure showing the average prevalence of Lassa fever confirmed cases across the different states within Nigeria.

RESPONSE: This has been done.

  1. COMMENT: The abbreviation CFR should be defined in Table 1. 

RESPONSE: This has been defined.

Reviewer 2 Report

Comments and Suggestions for Authors

This study presents a review of the epidemiology of Lassa fever in Nigeria. While the study is interesting, I have several concerns. I believe that if the authors can address these concerns, this study will add to the body of knowledge of lassa fever virus studies. 

Below are my comments

Major

  1. I find it hard to understand the contribution and novelty of this study. What is the major goal of this article?
  2. If this paper is truly a review, I expect the authors should present a PRISMA diagram (https://www.prisma-statement.org/) for their study. What are their search criteria? Why did they choose some articles and why was an article not chosen? What is the search strategy, etc.?
  3. What are the differences between this work and the studies in [33], [42], etc.? 

Minor

  1. There is a mistake on the last row of Table 1
  2. Work on spacing. See lines 35, 39, 65, 78, 113, 129, etc. 
  3. Align the font size of your manuscript. The font in lines 81-84 is different from others.
  4. In line 123, the authors writes 'A recent review ...' On looking at the citation, it is not a review paper.
  5. The keywords Lassa, fever, and Lassa fever virus are repetitions. It should just be Lassa fever virus

Author Response

RESPONSE TO REVIEWER TWO’S COMMENTS

MAJOR COMMENTS

  1. COMMENT: I find it hard to understand the contribution and novelty of this study. What is the major goal of this article?

RESPONSE: This study provides an overview of the trends in reference to the geo-spatial spread, increasing number of confirmed Lassa Fever cases and highlights effort by the national public health institute, Nigeria Centre for Disease Control (NCDC) to respond to the seasonal outbreak of Lassa fever in Nigeria. Specifically, the paper attempts to draw the attention of public health experts, epidemiologists, researchers and other stakeholders engaged in the control efforts of Lassa Fever of the need to focus on the reasons behind the expansion of the epidemic despite effort at control. Against this backdrop, surveillance approaches and control measures of Lassa Fever are highlighted as carried out within the framework of the Integrated Disease Surveillance and Response (IDSR) framework. 

The establishment of the Nigeria Centre for Disease Control (NCDC) and the adoption of the IDSR framework have significantly enhanced the country's capacity to detect outbreaks, monitor trends, and facilitate rapid responses in recent years. 

  1. COMMENT: If this paper is truly a review, I expect the authors should present a PRISMA diagram (https://www.prisma-statement.org/) for their study. What are their search criteria? Why did they choose some articles and why was an article not chosen? What is the search strategy, etc.?

RESPONSE: This paper was not originally intended to be a systematic review and hence we did not follow through with the protocols of conducting a systematic review. Rather, as highlighted above it was majorly intended to draw the attention of researchers, public health policy makers and other stakeholders to the increasing statistics on the morbidity, mortality including risk factors of Lassa fever. So, we are also proposing another title that best represent the goal of this paper:

  • Epidemiological Overview of Lassa Fever in Nigeria

  1. COMMENT: What are the differences between this work and the studies in [33], [42], etc.? 

RESPONSE: Thank you for raising this key point regarding the differences between our paper and the previously published studies on the topic. While earlier research addresses similar epidemiology themes, our study introduces new elements we believe contribute to advancing the discourse in this area.

Okwor et al. (2018) describe the epidemiological trends of Lassa fever in Nigeria from 1969 to 2017, while Dalhat et al. (2022) extended the analysis to the period from 2018 to 2022. Our study expands the timeline to include data up to 2024 and emphasizes the roles and approaches of the Integrated Disease Surveillance and Response (IDSR) team in early detection, reporting, analysis, and control across different levels of Nigeria’s healthcare system. This provides valuable insight into how the Nigeria Centre for Disease Control (NCDC) employs the IDSR framework to coordinate responses to Lassa fever outbreaks. 

Additionally, our review highlights the critical importance of interdisciplinary research as guided by the One Health Concept.

MINOR COMMENTS

  1. COMMENT: There is a mistake in the last row of Table 1

RESPONSE: It wasn’t a mistake, the report available for use on December 11 2024 when the author accessed the NCDC website stopped at week 46 instead of the end of the year (which should have been week 52). It’s updated now and displaying data for the whole year 2024.

  1. COMMENT: Work on spacing. See lines 35, 39, 65, 78, 113, 129, etc. 

RESPONSE: This has been corrected

  1. COMMENT: Align the font size of your manuscript. The font in lines 81-84 is different from others.

RESPONSE: Font size has been aligned

  1. COMMENT: In line 123, the author writes 'A recent review ...' On looking at the citation, it is not a review paper.

RESPONSE: The paper referred to was initially not cited, it has been included now both in the text and in the references.

  1. COMMENT: The keywords Lassa, fever, and Lassa fever virus are repetitions. It should just be Lassa fever virus

RESPONSE: This has been corrected. Thus, key words are now: Lassa fever virus; epidemiology; outbreaks; surveillance; response  

Round 2

Reviewer 1 Report

Comments and Suggestions for Authors

The manuscript underwent several beneficial improvements.

Minor comments:

  1. Please specify the LGA abbreviation in Table 1.
  2.  Authors need to indicate in the Figure caption if the numbers in Figure 3 and 4 correspond to the number of cases and why epidemiological week 52 was selected to show the prevalence of this virus across Nigeria.

Author Response

RESPONSE TO REVIEWER ONE’S COMMENTS

COMMENT: Minor comments:

  1. Please specify the LGA abbreviation in Table 1.
  2.  Authors need to indicate in the Figure caption if the numbers in Figure 3 and 4 correspond to the number of cases and why epidemiological week 52 was selected to show the prevalence of this virus across Nigeria.

RESPONSE: These have been effected in the manuscript.

Reviewer 2 Report

Comments and Suggestions for Authors

The authors' response to my first and third comments in the major comments must be incorporated in the introductory part of the article by clearly stating their goals and contributions. I can't find this in the revised version. The authors must state clearly how their manuscript is different from other articles published in this research area. I propose that this should be at the end of the introduction. 

Another important subtheme or perhaps an addition to their conclusion, is a suggestion for policymakers in order to mitigate the spread of Lassa fever. 

I am satisfied with the other revisions by the authors and I appreciate them for proposing to change the title of the article based on my comment. This article is an important epidemiological consideration, most especially in Nigeria and Africa. It will add value to the body of work in these research endeavors. 

Author Response

RESPONSE TO REVIEWER TWO’S COMMENTS

COMMENT: Comments and Suggestions for Authors

The authors' response to my first and third comments in the major comments must be incorporated in the introductory part of the article by clearly stating their goals and contributions. I can't find this in the revised version. The authors must state clearly how their manuscript is different from other articles published in this research area. I propose that this should be at the end of the introduction. 

Another important subtheme or perhaps an addition to their conclusion, is a suggestion for policymakers in order to mitigate the spread of Lassa fever. 

I am satisfied with the other revisions by the authors, and I appreciate them for proposing to change the title of the article based on my comment. This article is an important epidemiological consideration, most especially in Nigeria and Africa. It will add value to the body of work in these research endeavors. 

RESPONSE: These suggestions are very helpful and have been incorporated into the manuscript.